# MultiTask Learning for accelerated-MRI Reconstruction and Segmentation of Brain Lesions in Multiple Sclerosis

**Dimitrios Karkalousos**[1,2,3]                                    D.KARKALOUSOS@AMSTERDAMUMC.NL
**Ivana Išgum**[1,2,3,4]                                                  I.ISGUM@AMSTERDAMUMC.NL
**Henk A. Marquering**[1,2,3]                            H.A.MARQUERING@AMSTERDAMUMC.NL
**Matthan W.A. Caan**[1,3]                                  M.W.A.CAAN@AMSTERDAMUMC.NL

[1] *Department of Biomedical Engineering & Physics, Amsterdam University Medical Center, Location University of Amsterdam, Amsterdam, The Netherlands*

[2] *Department of Radiology & Nuclear Medicine, Amsterdam University Medical Center, Location University of Amsterdam, Amsterdam, The Netherlands*

[3] *Amsterdam Neuroscience, Brain Imaging, Amsterdam, The Netherlands*

[4] *Informatics Institute, University of Amsterdam, Amsterdam, The Netherlands*

**Editors:** Accepted for publication at MIDL 2023

## Abstract

This work proposes MultiTask Learning for accelerated-MRI Reconstruction and Segmentation (MTLRS). Unlike the common single-task approaches, MultiTask Learning identifies relations between multiple tasks to improve the performance of all tasks. The proposed MTLRS consists of a unique cascading architecture, where a recurrent reconstruction network and a segmentation network inform each other through hidden states. The features of the two networks are shared and implicitly enforced as inductive bias. To evaluate the benefit of MTLRS, we compare performing the two tasks of accelerated-MRI reconstruction and MRI segmentation with pre-trained, sequential, end-to-end, and joint approaches. A synthetic multicoil dataset is used to train, validate, and test all approaches with five-fold cross-validation. The dataset consists of 3D FLAIR brain data of relapsing-remitting Multiple Sclerosis patients with known white matter lesions. The acquisition is prospectively undersampled by approximately 7.5 times compared to clinical standards. Reconstruction performance is evaluated by Structural Similarity Index Measure (SSIM) and Peak Signal-to-Noise Ratio (PSNR). Segmentation performance is evaluated by Dice score for combined brain tissue and white matter lesion segmentation and by per lesion Dice score. Results show that MTLRS outperforms other evaluated approaches, providing high-quality reconstructions and accurate white matter lesion segmentation. A significant correlation was found between the performance of both tasks (SSIM and per lesion Dice score, $\rho = 0.92$, $p = 0.0005$). Our proposed MTLRS demonstrates that accelerated-MRI reconstruction and MRI segmentation can be effectively combined to improve performance on both tasks, potentially benefiting clinical settings.

**Keywords:** Multitask Learning, MRI, image reconstruction, segmentation, deep learning

## 1. Introduction

Acquisition, reconstruction, and analysis of Magnetic Resonance Imaging (MRI) are currently performed in a sequence of distinct tasks. Performing each task independently misses the opportunity to share valuable information between the tasks and jointly optimize their performance. MultiTask Learning (MTL) is a technique in which multiple domain-related

tasks are trained in parallel using shared features, effectively acting as inductive bias. MTL can implicitly identify task-relatedness, yielding improved generalization (Caruana, 1997). By utilizing the information in multiple tasks, the performance of each task can be improved. Recently task-adapted reconstruction was proposed to combine reconstruction with related tasks (Adler et al., 2022) in different approaches.

In a pre-trained approach, a reconstruction network and a segmentation network are trained separately to perform the tasks individually. In a sequential approach, the segmentation network is fine-tuned using the predictions of the reconstruction network. In an end-to-end approach, the reconstruction and the segmentation networks are trained together at the same time. For performing end-to-end accelerated-MRI reconstruction and MRI segmentation, Huang et al. (Huang et al., 2019) proposed the SEgmentation Recurrent Attention Network (SERANET), starting from the subsampled k-space to result in a segmentation. In a joint approach, the reconstruction and segmentation networks are trained end-to-end, computing a joint reconstruction and segmentation loss with a weighting factor balancing the two tasks. For performing the two tasks jointly, Sun et al. (Sun et al., 2019) proposed the SegNet, consisting of cascades of U-Nets for reconstruction and a separate decoder for segmentation, using the output of all the reconstruction encoders. Similarly, the Image Deep Structured Low-Rank (IDSLR) (Pramanik et al., 2021) and the RecSeg (Sui et al., 2021) methods perform joint reconstruction and segmentation. The IDSLR uses only the output of the final encoder for segmentation, while the RecSeg uses a second U-Net.

In this work, we formulate the inverse problem of accelerated-MRI reconstruction and the task of MRI segmentation as a multitask problem. In contrast to earlier methods, we show that performance on both tasks can be improved by informing each other through a recurrent approach. To this end, we leverage the Cascades of Independently Recurrent Inference Machines (CIRIM) (Karkalousos et al., 2022) for accelerated-MRI reconstruction and we add a segmentation network to the cascades to inform MultiTask Learning for accelerated-MRI Reconstruction and Segmentation (MTLRS). Following (Adler et al., 2022), the aim is to find a forward operator that directly maps accelerated-MR images to MRI segmentation. In MTLRS, this direct operator is modeled by coupling the output of the hidden layers of the reconstruction network with the output of the segmentation network. We develop and evaluate the proposed MLTRS using five-fold cross-validation on a synthetic multicoil dataset of 3D FLAIR data of relapsing-remitting Multiple Sclerosis patients with known white matter lesions.

## 2. Methods

### 2.1. MultiTask Learning for accelerated-MRI Reconstruction and Segmentation

The inverse problem of accelerated-MRI reconstruction can be formalized through a forward model. Let $x \in \mathbb{C}^n$ with $n = n_x \times n_y$, be a true image and let $y \in \mathbb{C}^m$, with $m << n$, be the set of sparse k-space measurements. The forward model describes $y$ as

$$y_i = A(x) + \sigma_i, i = 1, ..., c, \tag{1}$$

where $i$ denotes the current receiver coil, for a total of $c$ coils. $A : \mathbb{C}^n \mapsto \mathbb{C}^{n \times n_c}$ is the linear forward operator of accelerating MR acquisition, and $\sigma_i \in \mathbb{C}^n$ denotes the noise

from the scanner for the $i - th$ coil. $A$ is given by $A = U \odot \mathcal{F} \odot \epsilon$, where $U$ denotes the subsampling operator and $\mathcal{F}$ the Fourier transform. $\epsilon : \mathbb{C}^n \times \mathbb{C}^{n \times n_c} \mapsto \mathbb{C}^{n \times n_c}$ is the expand operator, transforming $x$ into $x_c$ multicoil images, given by $\epsilon(x) = (S_0 \odot x, ..., S_x \odot x) = (x_0, ..., x_c)$ where $S$ denote the coil sensitivity maps. Subsequently, the backward operator for projecting the sparse k-space to image space is given by $A^* = r \odot \mathcal{F}^{-1} \odot U^T$, where $\mathcal{F}^{-1}$ denotes the inverse Fourier transform. $r : \mathbb{C}^{n \times n_c} \times \mathbb{C}^{n \times n_c} \mapsto \mathbb{C}^n$ is the reduce operator computing a coil-combined image given by $r(x_0, ..., x_c) = \sum_{i=1}^{c} S_i^H \odot x_i$, where $H$ denotes the Hermitian complex conjugate.

When solving the inverse problem of accelerated-MRI reconstruction, the $y \mapsto x$ mapping (Eq. 1) can be found through a Maximum A Posteriori (MAP) estimation. Formulating the MAP estimation into a non-convex optimization scheme (Andrychowicz et al., 2016) results in updates of the form

$$x_{\iota+1} = x_\iota + \theta_\phi \left( \nabla_{y|x_\iota}, x_\iota \right), \tag{2}$$

at iteration $\iota$, for total number of iterations $I$. $\nabla_{y|x_\iota}$ is the gradient of the log-likelihood given by $\nabla_{y|x} := \frac{1}{\sigma^2} A^* \left( A(x) - y \right)$, assuming data are acquired under a Gaussian distribution. $\theta_\phi$ explicitly models the update rule using a Recurrent Neural Network (RNN).

Here, we use a learned inverse problem solver, the Cascades of Independently Recurrent Inference Machines (CIRIM) (Karkalousos et al., 2022). The update equations of the network for the first cascade are given by

$$
\begin{aligned}
h_0^{k=1} &= 0, & \hat{x}_0^{k=1} &= A^*(y), \\
h_{\iota+1}^{k=1} &= \theta_\phi^* \left( \nabla_{y|\hat{x}_\iota}, \hat{x}_\iota, h_\iota \right), & \hat{x_\iota}_{+1}^{k=1} &= \hat{x}_\iota + \theta_\phi \left( \nabla_{y|\hat{x}_\iota}, \hat{x}_\iota, h_{\iota+1} \right),
\end{aligned}
\tag{3}
$$

where $\theta_\phi^*$ is the updated model for the hidden state variable $h$ and $k$ denotes the current cascade, for total $K$ cascades. For the rest $2 \leq k \leq K$ cascades, we extend the CIRIM by including a segmentation network and further informing it of the segmentation task described by

$$s = T(x), \tag{4}$$

where $T : x \mapsto s$ is the generic forward segmentation operator and can be replaced by any segmentation network. MultiTask Learning for accelerated-MRI Reconstruction and Segmentation (MTLRS) is then realized by coupling the output of the hidden states with $s$, resulting in updates of the form

$$
\begin{aligned}
h_0^{k \geq 2} &= \hat{x}_I^{k-1} * s^{k-1}, & \hat{x}_0^{k \geq 2} &= \hat{x}_I^{k-1}, \\
h_{\iota+1}^{k \geq 2} &= \theta_\phi^* \left( \nabla_{y|\hat{x}_\iota^k}, \hat{x}_\iota^k, \hat{x}_I^{k-1} * s^{k-1} \right), & \hat{x_\iota}_{+1}^{k \geq 2} &= \hat{x}_\iota^k + \theta_\phi \left( \nabla_{y|\hat{x}_\iota^k}, \hat{x}_\iota^k, h_{\iota+1}^k \right).
\end{aligned}
\tag{5}
$$

In this way, the reconstruction informs the segmentation network and vice versa. A schematic representation is shown in Fig. 1.

## 2.2. Loss function

The loss function in MTLRS is described by a joint reconstruction $L_{recon}(\hat{x}, x)$ and segmentation $L_{seg}(\hat{s}, s)$ loss. The joint loss is given by

$$L^{joint} = \frac{1}{N} \sum_{n=1}^{N} (1 - \alpha) L_{recon}(\hat{x}_n, x_n) + \alpha L_{seg}(\hat{s}_n, s_n), \tag{6}$$

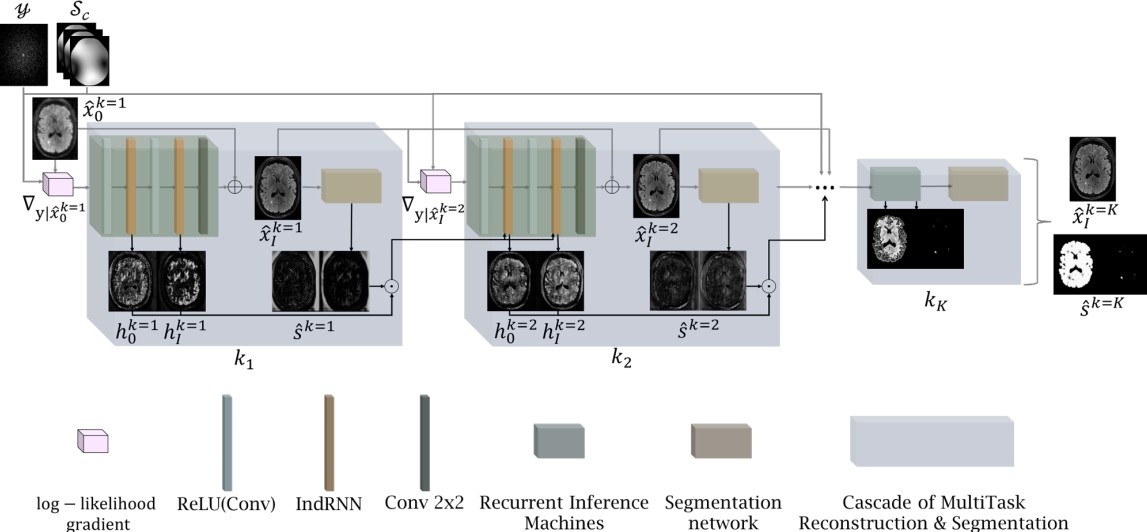

Figure 1: Schematic overview of the MultiTask Learning for accelerated-MRI Reconstruction and Segmentation (MTLRS) framework. MTLRS consists of $K$ cascades of a reconstruction network (top-leftmost block on each cascade) and a segmentation network (top-rightmost block on each cascade). On each cascade, the network first performs a reconstruction $(\hat{x_I}^k)$, next a segmentation $(\hat{s}^k)$, and finally couples the segmented output with the output of the hidden layers $(h_0^k$ and $h_I^k)$, to initialize the hidden layers of the next cascade. After $K$ cascades, the network outputs a final reconstruction $(\hat{x_I}^{k=K})$ and segmentation $(\hat{s}^{k=K})$ (top-rightmost).

where $n$ is the current batch and $N$ is the total number of training samples. $x$ is the ground truth image, $\hat{x}$ the predicted reconstruction, $s$ the ground truth segmentation label, and $\hat{s}$ the predicted segmentation. $\alpha$, with $0 \leq \alpha \leq 1$, is a weighting factor, balancing the influence of each task to the final loss.

$L^{recon}$ is usually computed on the magnitudes $x$ and $\hat{x}$, where $\hat{x}_0 = A(y)$ is the initially zero-filled reconstruction. In the case of the $l_1$-norm, the loss is given by

$$L^{l_1}(\hat{x}, x) = \frac{1}{N} \sum_{n=1}^{N} |\hat{x}_n - x_n|. \tag{7}$$

For MTLRS, $L^{recon}$ is weighted over the number of recurrent iterations. Thus Eq. 7 is reformulated as

$$L^{l_1}(\hat{x}, x) = \frac{1}{N} \sum_{n=1}^{N} \left( \frac{1}{qI} \sum_{\tau=1}^{I} w_\tau |\hat{x_{\tau n}} - x_n| \right), \tag{8}$$

where $q$ is the total number of pixels and $w_\tau$ is a vector containing $I$ weights, for a total number of iterations $I$, to emphasize the loss at later recurrent iterations. The weights are calculated as $w_\tau = 10^{-\frac{I-\tau}{I-1}}$.

For segmentation loss, we choose the commonly used binary cross-entropy loss and combine it with the Dice loss to ameliorate class imbalance given the very small size of

white matter lesions compared to segmented brain tissue. Therefore, a combined weighted binary cross-entropy and Dice loss assures stable loss computation. $L_{seg}$ is then given by

$$L_{seg}\left(\hat{s}, s\right) = \beta L^{CE}\left(\hat{s}, s\right) + (1 - \beta)L^{Dice}\left(\hat{s}, s\right), \tag{9}$$

where $L^{CE}\left(\hat{s}, s\right) = -\frac{1}{N}\sum_{n=1}^{N} s_n \ log \ \hat{s_n} + (1 - s_n) \ log \ (1 - \hat{s_n})$ and $L^{Dice}\left(\hat{s}, s\right) = 1 - \frac{2\sum_{n=1}^{N}\hat{s_n}s_n}{\sum_{n=1}^{N}\hat{s_n}^2 + \sum_{n=1}^{N}s_n^2}$. Finally, $\beta$ is a weighting factor balancing the contribution of each loss. In this work, we set $\beta = 0.5$.

## 2.3. Experiments

In our experiments, we evaluate the proposed MTLRS (Sec. 2.1) against other approaches which perform accelerated-MRI reconstruction and MRI segmentation without feature sharing. In a pre-trained approach, we train a reconstruction and a segmentation network separately and then use them independently at inference. In a sequential approach, we fine-tune the pre-trained segmentation network on the outputs of the reconstruction network. In an end-to-end approach, we train the two networks simultaneously but only compute a segmentation loss. In a joint approach, the two networks are trained with a joint reconstruction and segmentation loss (Eq. 6). The novelty of MTLRS lies in sharing features between the reconstruction and the segmentation network. Through a sequence of cascades, the segmented output is concatenated with the output of the hidden layers to initialize the hidden layers of the subsequent cascade. In that way, MTLRS is informed by the outputs of both tasks, in addition to a joint loss. In a joint approach, the network is only informed by the joint loss.

In all these approaches, we choose the Cascades of Independently Recurrent Inference Machines (CIRIM) as the reconstruction network and the Attention-UNet (Oktay et al., 2018) as the segmentation network, which empirically have been found to be well-performing models for each task. Additionally, we compare the performance of MTLRS with previously published methods. For this purpose, we implemented the end-to-end approach SEgmentation Recurrent Attention Network (SERANET) (Huang et al., 2019), and the joint approaches, RECSEGNET (Sui et al., 2021), Image Deep Structured Low-Rank (IDSLR) (Pramanik et al., 2021), and SEGNET (Sun et al., 2019). All models were trained and tested on an Nvidia Tesla V100 GPU with 32GB memory. The hyperparameter settings for all methods can be found in the Appendix. The code is publicly available at *****.

## 2.4. Dataset

A clinical dataset was used to train, validate, and test all methods using five-fold cross-validation. The dataset consisted of 3D FLAIR coil-combined magnitude brain images of 19 relapsing-remitting Multiple Sclerosis (MS) patients with white matter lesions. Data were acquired on a 3.0T scanner in our hospital. The local ethics review board approved this study, and the patients provided informed consent. Prospective undersampling was performed, accelerating imaging approximately 7.5 times under a Variable-Density Poisson disk distribution. Coil sensitivity maps were estimated using the caldir method of the BART toolbox (Uecker et al., 2015) on a fully-sampled reference.

The coil-combined magnitude images were used to synthesize multicoil complex data. To this end, we used a pre-trained CIRIM model trained only for reconstruction on 2D multislice FLAIR data (Muckley et al., 2021), accelerated approximately eight times under a Variable-Density Poisson disk distribution. Minimal random gaussian noise was added to the synthetic data, with a relative weighting factor of $10^-5$. Data were then retrospectively accelerated by approximately 7.5 times under a Variable-Density Poisson disk distribution. Next, we used the reconstructed images to predict two segmentation classes, brain tissue (combined white and gray matter) and white matter lesions, as a reference standard for MRI segmentation. To obtain brain tissue segmentations, we used the statistical parametric mapping (SPM) toolbox (Penny et al., 2007). To obtain white matter lesion segmentations, we used a pre-trained network for eye and tumor segmentation of retinoblastoma patients (Strijbis et al., 2021). All segmentations were visually inspected and manually corrected when necessary to assure segmentation accuracy.

## 2.5. Evaluation

For evaluating reconstruction, we compute Structural Similarity Index Measure (SSIM) (Wang et al., 2004) and Peak Signal-to-Noise-Ratio (PSNR) on the normalized magnitude images between the synthesized ground truth $x$ and the prediction $\hat{x}$. SSIM and PSNR are first computed per slice and per plane for each subject and then averaged to evaluate the reconstruction performance as a 3D volume. To evaluate segmentation, we calculate the Dice score as an overlap metric between the standard $s$ and the prediction $\hat{s}$. Dice score is reported for the combined (white and gray matter) tissue and white matter lesion segmentation and for only the white matter lesion segmentation. Dice scores are computed across all planes and slices for all subjects. To assess whether a correlation in performance between both tasks exists, we correlated SSIM and per lesion Dice scores using Spearman's rank test.

## 3. Results

Figure 2 shows an overall comparison, averaged over five-folds, of MTLRS to the Pre-Trained, Sequential, End-to-End, and Joint approaches. Note that the Sequential and the End-to-End approaches are optimized only for segmentation. MTLRS performed best on both reconstruction and segmentation. The Joint approach performed close to MTLRS but with a larger standard deviation. The Pre-Trained approach dropped in performance on both tasks, while it performed on par with the Sequential approach on segmentation, showing no apparent benefit when further optimizing the segmentation model on the reconstructed outputs. The End-to-End approach was the worst segmentation method, indicating the need for a joint loss rather than only segmentation loss.

Table 1 reports the performance of MTLRS and the evaluated previously published methods, averaged over five-folds. In both tasks, MTLRS outperformed the RECSEGNET, IDSLR, SEGNET, and SERANET, showing a clear advantage for the multitask approach. Overview tables reporting the performance of all approaches and previously published methods on each fold can be found in the Appendix.

Figure 3 shows an example of a reconstructed and segmented axial slice by MTLRS and the evaluated previously published methods. MTLRS provided the highest reconstruction

quality (SSIM) and the most accurate lesion segmentation (Dice). The RECSEGNET performed comparably with MTLRS in reconstruction, while the IDSLR and SEGNET reduced reconstruction performance further. The SERANET was the worst-performing method on reconstruction.

SSIM and lesions Dice scores were significantly correlated ($\rho = 0.92$, $p = 0.0005$). More examples of reconstructions and segmentations can be found in the Appendix.

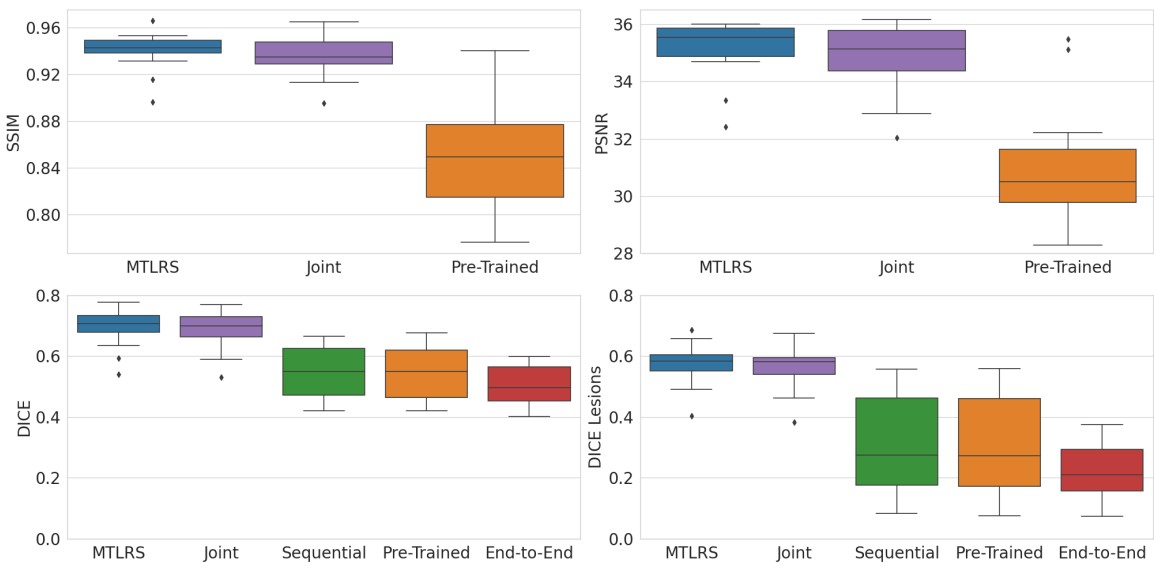

Figure 2: Quantitative evaluation averaged over five-folds when performing accelerated-MRI reconstruction and MRI segmentation with different approaches (x-axis). Data were retrospectively undersampled 7.5 times. SSIM and PSNR (top) evaluate reconstruction. DICE and DICE Lesions (bottom) evaluate segmentation.

Table 1: Overall comparison, averaged over five-folds, of MTLRS to previously published methods when performing accelerated-MRI reconstruction and MRI segmentation. SSIM and PSNR evaluate reconstruction. DICE and DICE Lesions evaluate segmentation. Metrics are computed on retrospectively undersampled data by 7.5 times. The arrow pointing upward indicates higher is better. Methods are sorted by DICE, while the best-performing method is shown in bold.

| Method | SSIM ↑ | PSNR ↑ | DICE ↑ | DICE Lesions ↑ |
|---|---|---|---|---|
| MTLRS | **0.940 ± 0.017** | **35.26 ± 1.30** | **0.691 ± 0.065** | **0.574 ± 0.069** |
| RECSEGNET | 0.787 ± 0.041 | 28.93 ± 0.99 | 0.512 ± 0.059 | 0.229 ± 0.086 |
| SERANET | | | 0.508 ± 0.063 | 0.221 ± 0.082 |
| IDSLR | 0.758 ± 0.034 | 27.31 ± 0.96 | 0.490 ± 0.054 | 0.186 ± 0.075 |
| SEGNET | 0.749 ± 0.039 | 27.07 ± 1.14 | 0.479 ± 0.056 | 0.178 ± 0.065 |

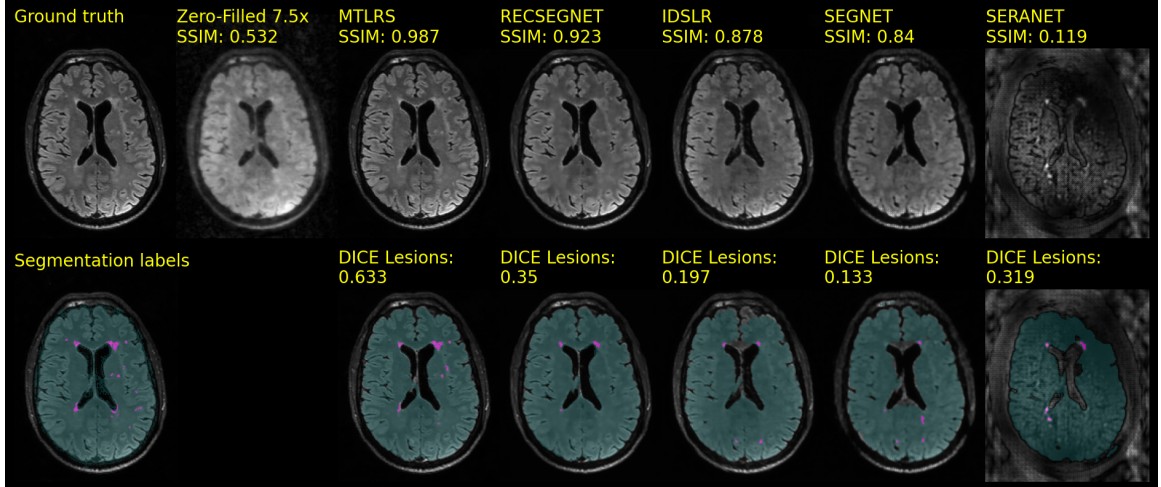

Figure 3: Reconstruction and segmentation of an axial slice with white matter lesions. An acceleration factor of approximately 7.5 was used to undersample the data retrospectively (top-second column). Methods are sorted by SSIM. SSIM is computed for evaluating reconstruction performance against the ground truth (top-first column). The per lesions Dice score is computed to evaluate segmentation performance against the reference labels (bottom-first column).

## 4. Discussion & Conclusion

We proposed MultiTask Learning for accelerated-MRI Reconstruction and Segmentation (MTLRS). MultiTask Learning was realized through a unique cascading network architecture consisting of a recurrent reconstruction network and segmentation network. The output of the hidden layers was combined with the segmented images to inform a sequence of cascades, thus serving as an inductive bias. Performance was evaluated using five-fold cross-validation. MTLRS outperformed the Pre-Trained, Sequential, and End-to-End approaches and existing methods (RECSEGNET, SERANET, IDSLR, SEGNET) on reconstructing 7.5 times accelerated 3D FLAIR brain data of Multiple Sclerosis patients and on segmenting white matter lesions identified on this data. Additionally, it improved marginally upon the Joint approach. The reason could lie in the fact that the reconstruction network architecture used in MTLRS and the joint and pre-trained approaches was the same as the pre-trained network used in synthesizing the multicoil dataset. Therefore, future work will evaluate our method on a dataset where fully sampled reference data is available, e.g., knee data from the recently held KS-challenge (Bharadwaj et al.). Interestingly, a strong correlation was found between the quality metrics of both tasks. The results suggest that improved dealiasing during reconstruction leads to improved contrast and better-defined lesion boundaries, thereby supporting a more accurate segmentation. In future work, more tasks can be combined, such as classifying the underlying pathologies and improving performance by informing each other. Thus, MultiTask Learning is yet to be further explored, with potentially a high value if applied in the clinical setting, where aside from improving performance, the need for waiting time between multiple tasks would not be needed.

## Acknowledgments

This publication is based on the STAIRS project under the TKI-PPP program. The collaboration project is co-funded by the PPP Allowance made available by Health Holland, Top Sector Life Sciences & Health, to stimulate public-private partnerships.

M.W.A. Caan is shareholder of Nico.Lab International Ltd.

We want to thank Hanneke E. Hulst for providing the dataset of the Multiple-Sclerosis patients, Samantha Noteboom for helping with the initial segmentation of the data, and Lysander de Jong for his contribution to assessing the reconstruction and segmentation as a joint problem.

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

## Appendix

### Hyperparameters

In our experiments, we set the hyperparameters of the related compared work according to what is reported by the authors in the original work.

For the CIRIM, we set the number of features to 64 for the convolutional and recurrent layers, cascades to 5, and recurrent iterations to 8. For the AttentionUNet, we set the number of features to 64, pooling layers to 2, and dropout to 0. For the SERANET, we chose the U-Net as the reconstruction network. We set the number of features to 32, pooling layers to 4, and dropout to 0 for the reconstruction, segmentation, and recurrent modules, and built three reconstruction blocks. For the RECSEGNET and the IDSLR, we set the number of features to 64, pooling layers to 2, and dropout to 0. For the IDSLR, the number of iterations was set to 5. $\alpha$ was set to 0.5 for the RECSEGNET and $10e-6$ for IDSLR. Finally, for the SEGNET, we set the number of features to 64, pooling layers to 2, dropout to 0, and cascades to 5.

For finding the optimal value for $\alpha$ in the joint loss for MTLRS (Eq. 6), we performed a hyperparameter search as presented in Figure 4. The tested values are $0.01, 0.1, 0.5, 0.9, 0.99$, going from favoring the reconstruction loss to balancing the loss to favoring the segmentation loss. The optimal $\alpha$ value was found to be 0.9.

We used ADAM as optimizer for all methods and set the learning rate to $10^{-4}$.

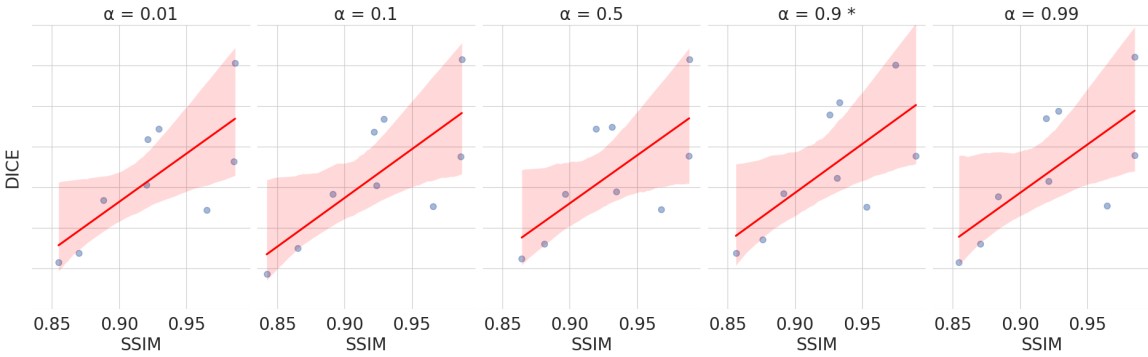

Figure 4: Hyperparameter search for finding the optimal $\alpha$ value in the joint loss (Eq. 6). Reconstruction and segmentation performance are realized on an SSIM (x-axis) over Dice (y-axis) plot. From left to right. The $*$ indicates the value resulting in the best SSIM & Dice scores.

### Overview five-fold cross-validation

Tables 2 and 3 report performance on each of the five folds of the cross-validation, of MTLRS and all compared approaches and previously published methods when performing accelerated-MRI reconstruction and MRI segmentation. MTLRS was the best-performing method overall on all folds and all metrics. Only on the second fold and on PSNR the Joint approach scored higher than MTLRS.

Table 2: Overall performance of MTLRS and the Pre-Trained, Sequential, End-to-End, and Joint approaches for five-fold cross-validation when performing accelerated-MRI reconstruction and MRI segmentation. SSIM and PSNR evaluate reconstruction. DICE and DICE Lesions evaluate segmentation. Metrics are computed on retrospectively undersampled data by 7.5 times. The arrow pointing upward indicates higher is better. Methods are sorted by DICE, while the best-performing method is shown in bold.

| Method | SSIM ↑ | PSNR ↑ | Dice ↑ | Dice Lesions ↑ |
|---|---|---|---|---|
| | | Fold 1 | | |
| MTLRS | **0.936 ± 0.036** | **34.94 ± 3.50** | **0.656 ± 0.094** | **0.511 ± 0.098** |
| Joint | **0.936 ± 0.037** | 34.79 ± 3.46 | 0.651 ± 0.096 | 0.505 ± 0.105 |
| Pre-Trained | 0.821 ± 0.090 | 29.84 ± 3.33 | 0.475 ± 0.111 | 0.197 ± 0.172 |
| End-to-End | | | 0.474 ± 0.075 | 0.178 ± 0.110 |
| Sequential | | | 0.473 ± 0.112 | 0.178 ± 0.110 |
| | | Fold 2 | | |
| MTLRS | **0.961 ± 0.025** | 36.69 ± 2.75 | **0.706 ± 0.074** | **0.588 ± 0.091** |
| Joint | 0.959 ± 0.034 | **37.13 ± 3.59** | 0.704 ± 0.073 | 0.587 ± 0.091 |
| Sequential | | | 0.604 ± 0.150 | 0.407 ± 0.270 |
| Pre-Trained | 0.882 ± 0.075 | 31.43 ± 2.85 | 0.601 ± 0.151 | 0.400 ± 0.272 |
| End-to-End | | | 0.555 ± 0.138 | 0.319 ± 0.236 |
| | | Fold 3 | | |
| MTLRS | **0.944 ± 0.027** | **35.72 ± 3.20** | **0.677 ± 0.090** | **0.558 ± 0.107** |
| Joint | 0.933 ± 0.033 | 34.96 ± 3.20 | 0.664 ± 0.095 | 0.528 ± 0.120 |
| Pre-Trained | 0.838 ± 0.067 | 30.65 ± 2.80 | 0.480 ± 0.147 | 0.211 ± 0.246 |
| Sequential | | | 0.489 ± 0.141 | 0.220 ± 0.238 |
| End-to-End | | | 0.487 ± 0.111 | 0.204 ± 0.180 |
| | | Fold 4 | | |
| MTLRS | **0.940 ± 0.032** | **35.26 ± 3.40** | **0.707 ± 0.051** | **0.572 ± 0.070** |
| Joint | 0.937 ± 0.035 | 34.97 ± 3.67 | 0.697 ± 0.059 | 0.552 ± 0.080 |
| Sequential | | | 0.506 ± 0.093 | 0.222 ± 0.184 |
| Pre-Trained | 0.814 ± 0.080 | 29.63 ± 2.95 | 0.500 ± 0.094 | 0.219 ± 0.182 |
| End-to-End | | | 0.495 ± 0.083 | 0.201 ± 0.155 |
| | | Fold 5 | | |
| MTLRS | **0.923 ± 0.068** | **34.18 ± 4.57** | **0.654 ± 0.066** | **0.570 ± 0.085** |
| Pre-Trained | 0.918 ± 0.064 | 33.82 ± 4.52 | 0.636 ± 0.087 | 0.540 ± 0.112 |
| Joint | 0.915 ± 0.061 | 33.64 ± 4.52 | 0.646 ± 0.070 | 0.562 ± 0.082 |
| Sequential | | | 0.634 ± 0.080 | 0.542 ± 0.114 |
| End-to-End | | | 0.490 ± 0.104 | 0.233 ± 0.153 |

Table 3: Overall performance of MTLRS and previously published methods for five-fold cross-validation when performing accelerated-MRI reconstruction and MRI segmentation. SSIM and PSNR evaluate reconstruction. DICE and DICE Lesions evaluate segmentation. Metrics are computed on retrospectively undersampled data by 7.5 times. The arrow pointing upward indicates higher is better. Methods are sorted by DICE, while the best-performing method is shown in bold

| Method | SSIM ↑ | PSNR ↑ | Dice ↑ | Dice Lesions ↑ |
|---|---|---|---|---|
| | | Fold 1 | | |
| MTLRS | **0.936 ± 0.036** | **34.94 ± 3.50** | **0.656 ± 0.094** | **0.511 ± 0.098** |
| RECSEGNET | 0.781 ± 0.088 | 28.61 ± 2.51 | 0.481 ± 0.081 | 0.175 ± 0.120 |
| SERANET | | | 0.472 ± 0.063 | 0.160 ± 0.076 |
| SEGNET | 0.761 ± 0.072 | 27.47 ± 2.10 | 0.457 ± 0.066 | 0.129 ± 0.074 |
| IDSLR | 0.760 ± 0.075 | 27.48 ± 1.97 | 0.457 ± 0.067 | 0.129 ± 0.067 |
| | | Fold 2 | | |
| MTLRS | **0.961 ± 0.025** | **36.69± 2.75** | **0.706 ± 0.074** | **0.588 ± 0.091** |
| RECSEGNET | 0.850 ± 0.079 | 30.29 ± 2.41 | 0.568 ± 0.128 | 0.322 ± 0.238 |
| SERANET | | | 0.548 ± 0.109 | 0.280 ± 0.200 |
| IDSLR | 0.808 ± 0.063 | 27.51 ± 2.24 | 0.526 ± 0.111 | 0.242 ± 0.204 |
| SEGNET | 0.791 ± 0.068 | 27.03 ± 2.33 | 0.505 ± 0.091 | 0.213 ± 0.150 |
| | | Fold 3 | | |
| MTLRS | **0.944 ± 0.027** | **35.72 ± 3.20** | **0.677 ± 0.090** | **0.558 ± 0.107** |
| SERANET | | | 0.495 ± 0.096 | 0.202 ± 0.151 |
| RECSEGNET | 0.792 ± 0.072 | 29.40 ± 2.20 | 0.492 ± 0.110 | 0.199 ± 0.185 |
| SEGNET | 0.756 ± 0.067 | 27.86 ± 1.76 | 0.462 ± 0.096 | 0.157 ± 0.139 |
| IDSLR | 0.755 ± 0.068 | 27.83 ± 1.83 | 0.455 ± 0.093 | 0.137 ± 0.129 |
| | | Fold 4 | | |
| MTLRS | **0.940 ± 0.032** | **35.26 ± 3.40** | **0.707 ± 0.051** | **0.572 ± 0.070** |
| SERANET | | | 0.526 ± 0.049 | 0.226 ± 0.095 |
| RECSEGNET | 0.773 ± 0.080 | 28.58 ± 2.33 | 0.493 ± 0.059 | 0.186 ± 0.118 |
| SEGNET | 0.747 ± 0.067 | 27.46 ± 2.01 | 0.477 ± 0.038 | 0.150 ± 0.071 |
| IDSLR | 0.744 ± 0.075 | 27.46 ± 2.04 | 0.480 ± 0.047 | 0.147 ± 0.096 |
| | | Fold 5 | | |
| MTLRS | **0.923 ± 0.068** | **34.18 ± 4.57** | **0.654 ± 0.066** | **0.570 ± 0.085** |
| RECSEGNET | 0.749 ± 0.122 | 27.94 ± 3.21 | 0.507 ± 0.120 | 0.268 ± 0.205 |
| IDSLR | 0.719 ± 0.107 | 26.24 ± 2.32 | 0.492 ± 0.110 | 0.243 ± 0.188 |
| SERANET | | | 0.490 ± 0.104 | 0.233 ± 0.153 |
| SEGNET | 0.690 ± 0.103 | 25.48 ± 2.26 | 0.454 ± 0.111 | 0.198 ± 0.164 |

**Overview examples**

Figures 5 and 6 show examples of reconstructed and segmented slices of the coronal and sagittal view, with white matter lesions identified in all slices. An acceleration factor of approximately 7.5 was used to undersample the data retrospectively. MTLRS provided the highest reconstruction quality (SSIM) and the most accurate lesions segmentation (Dice) in all cases. The RECSEGNET dropped significantly both in SSIM and Dice score by oversimplifying the reconstruction and slightly overestimating lesion volume. The same behavior is observed by the IDSLR and the SEGNET, reducing performance further. The SERANET performed poorly on reconstruction, while segmentation performance was comparable or better to the RECSEGNET, IDSLR, and SEGNET.

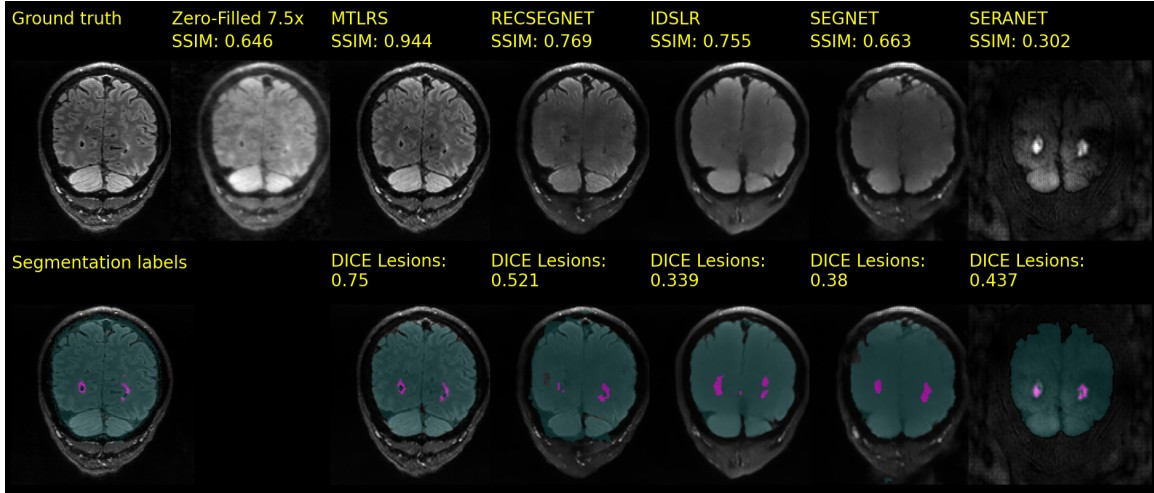

Figure 5: Reconstruction and segmentation of a coronal slice with white matter lesions. An acceleration factor of approximately 7.5 was used to undersample the data retrospectively (top-second column). Methods are sorted by SSIM. SSIM is computed for evaluating reconstruction performance against the ground truth (top-first column). The per lesions Dice score is computed to evaluate segmentation performance against the reference labels (bottom-first column).

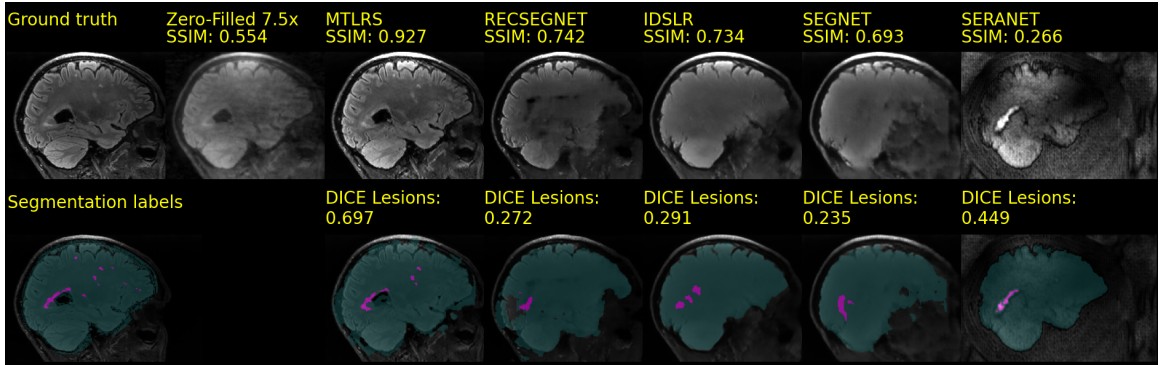

Figure 6: Reconstruction and segmentation of a sagittal slice with white matter lesions. An acceleration factor of approximately 7.5 was used to undersample the data retrospectively (top-second column). Methods are sorted by SSIM. SSIM is computed for evaluating reconstruction performance against the ground truth (top-first column). The per lesions Dice score is computed to evaluate segmentation performance against the reference labels (bottom-first column).

