# OpenReview forum: "MultiTask Learning for accelerated-MRI Reconstruction and Segmentation of Brain Lesions in Multiple Sclerosis"
_MIDL.io/2023/Conference — MIDL 2023 Poster_

### Official Review · Reviewer_ecCw · 2023-02-03

**Confidence:** 3
**Preliminary Rating:** 4
**Recommendation:** Poster

**Summary:**

The authors propose a method for simultaneous learning of reconstruction and segmentation of MRI images. The method consists of a recurrent reconstruction network and a segmentation network that share features during the training process. The method is tested in a synthetic dataset of 3D FLAIR brain data obtained from remitting multiple sclerosis patients with white matter lesions. Results show that the images reconstructed with the proposed method have higher similarity to the reference standard and segment the lesions better than other sequential or even joint training approaches.

**Strengths:**

The paper is interesting and relevant to the current state of the art. The proposed multi-task learning outperformed sequential, pre-trained and end-to-end methods. Segmentation of lesions is better than other methods by a large margin.

**Weaknesses:**

Evaluation: as the authors mention, they split their 19 patients into 13 for training, 3 for validation and three for testing. Unless there is a misunderstanding on my end, this is a very small dataset for testing. At least n-fold cross validation should be performed.
The paper is hard to follow.


**Deanonymize Review:**

no

**Paper Type:**

methodological development

**Questions To Address In The Rebuttal:**

Figure 1 could include a larger explanation of the network in the caption.
Equation 11 seems wrong, there are two beta symbols, isn’t it one beta and the other (1-beta)
Why do the authors use a combination of DICE and BCE for the segmentation loss function?

---

### Official Review · Reviewer_7Xo4 · 2023-02-04

**Confidence:** 4
**Preliminary Rating:** 3

**Summary:**

The authors use a multi task learning paradigm to simultaneously train a magnetic resonance imaging (MRI) reconstruction model and segmentation model. The authors extend the work (Karkalousos et al., 2022), which introduced the Cascades of Independently Recurrent Inference Machine to reconstruct MRI images from k space, to a model that is also capable of performing segmentation.

**Strengths:**

The proposed method appears to perform well in the benchmarks, outperforming other methods in reconstruction and segmentation metrics consistently. Additionally, the proposed method appears to be significantly better than the current state of the art.

**Weaknesses:**

The authors over complicate matters by using a lot of mathematical notation that often doesn't help, and at times hinders, the flow of the paper, making some concepts unnecessarily difficult to understand. I novelty of adding a segmentation loss to an existing model is limited.

**Deanonymize Review:**

no

**Detailed Comments:**

- "wτ a weighting vector of length I" - this actually a scalar, I believe you mean there are I of these.

**Paper Type:**

methodological development

**Questions To Address In The Rebuttal:**

- Equation nine needs to be reviewed, there is no use of the index of summation which then makes it simplify to |\hat{x}-x|
- Equation eleven also needs to be review, one of the beta terms should be (1-beta)
- Please elaborate more the other models/architectures in the experiments sections, specifically, what is the difference between the joint model and the proposed model?
- Are all results in table 1 on the same dataset?

---

### Official Review · Reviewer_RBFY · 2023-02-06

**Confidence:** 3
**Preliminary Rating:** 3
**Recommendation:** Poster

**Summary:**

This paper proposed a multi-task framework for joint MRI reconstruction and segmentation, where a recurrent reconstruction and segmentation networks share hidden layers. The proposed methods were compared with different training approaches and evaluated on 3D FLAIR multiple sclerosis patients dataset with white matter lesions. It was demonstrated that the proposed methods outperforms other approaches.

**Strengths:**

This paper is overall well written and easy to follow. As shown in previous studies, multi-task learning leverage the performance of each task when the features of multiple output tasks are highly correlated. This paper coincides with these previous findings.

**Weaknesses:**

The sample sizes are limited. Also, considering such a small number of samples, the evaluation should be performed in a cross-validation. It’s hard to make a conclusion based on the results of three test subjects. Similarly, it’s difficult to say whether the improvement of the proposed method is statistically significant.

**Deanonymize Review:**

no

**Detailed Comments:**

Please double check the notations in section 2.1. Does the index ‘i’ start from 0 or 1? Please clarify. The equation (2) for MAP should be written in math symbol or italic.

**Paper Type:**

methodological development

**Questions To Address In The Rebuttal:**

-	It seems that the ground truth segmentation was obtained using the statistical parametric mapping toolbox and pre-trained network from the other paper. Did the authors perform any further manual correction? How much reliable was this automatic segmentation?
-	Can the authors elaborate more on joint approach compared to the proposed MTLRS? What is the difference between them?

---

### Meta-Review · Area_Chair_z1xR · 2023-02-23

**Recommendation:** Accept (Poster)
**Confidence:** 4

**Metareview:**

This paper introduces a new method to perform accelerated MR reconstruction and segmentation via multi-task learning. The main concern raised by the reviewers is the dataset size and the clarity of the mathematical formulation. As requested by the reviewers, the authors have now included a 5-fold cross-validation scheme to improve the experimental results. They have also simplified the mathematical notation when possible. After reading the reviewers comments and the rebuttal, I lean towards acceptance of this work.